# Long-Term Retention Rate of Tofacitinib in Rheumatoid Arthritis: An Italian Multicenter Retrospective Cohort Study

**DOI:** 10.3390/medicina59081480

**Published:** 2023-08-17

**Authors:** Marino Paroli, Andrea Becciolini, Elena Bravi, Romina Andracco, Valeria Nucera, Simone Parisi, Francesca Ometto, Federica Lumetti, Antonella Farina, Patrizia Del Medico, Matteo Colina, Alberto Lo Gullo, Viviana Ravagnani, Palma Scolieri, Maddalena Larosa, Marta Priora, Elisa Visalli, Olga Addimanda, Rosetta Vitetta, Alessandro Volpe, Alessandra Bezzi, Francesco Girelli, Aldo Biagio Molica Colella, Rosalba Caccavale, Eleonora Di Donato, Giuditta Adorni, Daniele Santilli, Gianluca Lucchini, Eugenio Arrigoni, Ilaria Platè, Natalia Mansueto, Aurora Ianniello, Enrico Fusaro, Maria Chiara Ditto, Vincenzo Bruzzese, Dario Camellino, Gerolamo Bianchi, Francesca Serale, Rosario Foti, Giorgio Amato, Francesco De Lucia, Ylenia Dal Bosco, Roberta Foti, Massimo Reta, Alessia Fiorenza, Guido Rovera, Antonio Marchetta, Maria Cristina Focherini, Fabio Mascella, Simone Bernardi, Gilda Sandri, Dilia Giuggioli, Carlo Salvarani, Veronica Franchina, Francesco Molica Colella, Giulio Ferrero, Alarico Ariani

**Affiliations:** 1Department of Clinical, Internist, Anesthesiologic and Cardiovascular Sciences, Sapienza University of Rome, 00185 Rome, Italy; rosalba_caccavale@yahoo.it; 2Internal Medicine and Rheumatology Unit, University Hospital of Parma, 43126 Parma, Italy; beccio@yahoo.it (A.B.); eleonoradidonato@ymail.com (E.D.D.); gadorni@ao.pr.it (G.A.); dsantilli@ao.pr.it (D.S.); glucchini@ao.pr.it (G.L.); dott.alaricoariani@libero.it (A.A.); 3Rheumatology Unit, Guglielmo da Saliceto Hospital, 29121 Piacenza, Italy; e.bravi@ausl.pc.it (E.B.); e.arrigoni@ausl.pc.it (E.A.); i.plate@ausl.pc.it (I.P.); 4Internal Medicine Unit, Imperia Hospital, 18100 Imperia, Italy; r.andracco@gmail.com (R.A.); natalia.mansueto@libero.it (N.M.); 5Rheumatology Unit, ASL Novara, 28100 Novara, Italy; v.nucera@asl.novara.it (V.N.); a.ianniello@asl.novara.it (A.I.); 6Rheumatology Department, Azienda Ospedaliero-Universitaria Città della Salute e della Scienza di Torino, 10126 Torino, Italy; simone.parisi@hotmail.it (S.P.); fusaro.reumatorino@gmail.com (E.F.); mariachiaraditto@gmail.com (M.C.D.); 7Rheumatology Unit, Azienda ULSS 6 Euganea, 35131 Padova, Italy; f.ometto@gmail.com; 8Rheumatology Unit, Azienda USL of Modena and AOU Policlinico of Modena, 41100 Modena, Italy; fedelumetti@gmail.com; 9Internal Medicine Unit, Augusto Murri Hospital, 63900 Fermo, Italy; antonella_farina@hotmail.com; 10Internal Medicine Unit, Civitanova Marche Hospital, 62012 Civitanova Marche, Italy; patdelmedico@hotmail.com; 11Rheumatology Unit, Internal Medicine Division, Department of Medicine and Oncology, Santa Maria della Scaletta Hospital, 40026 Imola, Italy; matteo.colina2@unibo.it; 12Rheumatology Unit, Alma Mater Studiorum—University of Bologna, 40126 Bologna, Italy; 13Rheumatology Unit, ARNAS Garibaldi, 95124 Catania, Italy; albertologullo@virgilio.it; 14Rheumatology Unit, Santa Chiara Hospital APSS—Trento, 38122 Trento, Italy; viviana.ravagnani@gmail.com; 15Rheumatology Unit, Nuovo Regina Margherita Hospital, 00154 Roma, Italy; palma.scolieri@gmail.com (P.S.); vinbruzzese@tiscali.it (V.B.); 16Division of Rheumatology, Department of Medical Specialties, Azienda Sanitaria Locale 3 Genovese, 16132 Genova, Italy; madlarosa8@gmail.com (M.L.); dario.camellino@asl3.liguria.it (D.C.); gerolamo.bianchi@asl3.liguria.it (G.B.); 17Rheumatology Unit, ASL CN1, 12100 Cuneo, Italy; marta.priora@gmail.com (M.P.); francesca.serale@gmail.com (F.S.); 18Rheumatology Unit, Policlinico San Marco Hospital, 95121 Catania, Italy; elivisa21@gmail.com (E.V.); rosfoti5@gmail.com (R.F.); giorgioamato@hotmail.it (G.A.); francescodelucia89@yahoo.it (F.D.L.); yleniadalbosco@gmail.com (Y.D.B.); robertafoti@hotmail.com (R.F.); 19Rheumatology Unit, AUSL of Bologna—Policlinico Sant’Orsola—AOU—IRCCS of Bologna, 40138 Bologna, Italy; olga.addimanda@ausl.bologna.it (O.A.); massimo.reta@ausl.bologna.it (M.R.); 20Unit of Rheumatology, ASL VC Sant’ Andrea Hospital, 13100 Vercelli, Italy; rosetta.vitetta@aslvc.piemonte.it (R.V.); alessia.fiorenza@aslvc.piemonte.it (A.F.);; 21Unit of Rheumatology, IRCCS Sacro Cuore Don Calabria Hospital, 37024 Negrar di Valpolicella, Italy; avolpe127@gmail.com (A.V.); antonio.marchetta@sacrocuore.it (A.M.); 22Internal Medicine and Rheumatology Unit, AUSL della Romagna—Rimini, 47924 Rimini, Italy; alessandra.bezzi@auslromagna.it (A.B.); mariacristina.focherini@auslromagna.it (M.C.F.); fabio.mascella@auslromagna.it (F.M.); 23Rheumatology Unit, G.B. Morgagni—L. Pierantoni Hospital, 47121 Forlì, Italy; francesco.girelli@auslromagna.it (F.G.); siiberna@yahoo.it (S.B.); 24Rheumatology Unit, Division of Internal Medicine, Azienda Ospedaliera Papardo, 98158 Messina, Italy; aldomolica@alice.it; 25Rheumatology Unit, University of Modena and Reggio Emilia, 41125 Modena, Italy; gilda.sandri@unimore.it (G.S.); dilia.giuggioli@unimore.it (D.G.); carlo.salvarani@unimore.it (C.S.); 26Medical Oncology Unit, Azienda Ospedaliera Papardo, 98158 Messina, Italy; verifra82@yahoo.it; 27Medicine Unit, Milano-Bicocca University, 20126 Milano, Italy; francesco.molica3@gmail.com; 28Unit of Diagnostic and Interventional Radiology, Santa Corona Hospital, 17027 Pietra Ligure, Italy; giulio.ferrero@gmail.com

**Keywords:** tofacitinib, Janus kinase inhibitors, rheumatoid arthritis, drug retention rate

## Abstract

*Background*: Tofacitinib (TOFA) was the first Janus kinase inhibitor (JAKi) to be approved for the treatment of rheumatoid arthritis (RA). However, data on the retention rate of TOFA therapy are still far from definitive. *Objective*: The goal of this study is to add new real-world data on the TOFA retention rate in a cohort of RA patients followed for a long period of time. *Methods*: A multicenter retrospective study of RA subjects treated with TOFA as monotherapy or in combination with conventional synthetic disease-modifying antirheumatic drugs (csDMARDs) was conducted in 23 Italian tertiary rheumatology centers. The study considered a treatment period of up to 48 months for all included patients. The TOFA retention rate was assessed with the Kaplan–Meier method. Hazard ratios (HRs) for TOFA discontinuation were obtained using Cox regression analysis. *Results*: We enrolled a total of 213 patients. Data analysis revealed that the TOFA retention rate was 86.5% (95% CI: 81.8–91.5%) at month 12, 78.8% (95% CI: 78.8–85.2%) at month 24, 63.8% (95% CI: 55.1–73.8%) at month 36, and 59.9% (95% CI: 55.1–73.8%) at month 48 after starting treatment. None of the factors analyzed, including the number of previous treatments received, disease activity or duration, presence of rheumatoid factor and/or anti-citrullinated protein antibody, and presence of comorbidities, were predictive of the TOFA retention rate. Safety data were comparable to those reported in the registration studies. *Conclusions*: TOFA demonstrated a long retention rate in RA in a real-world setting. This result, together with the safety data obtained, underscores that TOFA is a viable alternative for patients who have failed treatment with csDMARD and/or biologic DMARDs (bDMARDs). Further large, long-term observational studies are urgently needed to confirm these results.

## 1. Introduction

Rheumatoid arthritis (RA) is a chronic inflammatory rheumatic disease that affects both small and large joints. If left untreated, rheumatoid arthritis can cause joint erosions and deformities that can lead to severe disability. In addition, extra-articular manifestations of RA, such as interstitial lung disease, vasculitis, and lymphoma, can be very serious and even life-threatening [1]. Therefore, it is necessary to interrupt the inflammatory process to prevent the extension of the damage and allow an acceptable quality of life for the patient.

Although the introduction of conventional synthetic disease-modifying antirheumatic drugs (csDMARDs) and later biologic DMARDs (bDMARDs) into therapy has radically changed the prognosis of RA, many patients respond unsatisfactorily to these therapies [2,3]. It should also be noted that available DMARDs often prove ineffective for achieving and maintaining stable disease remission in most patients. In this regard, new approaches, including the use of nanostructures, offer promising possibilities for improved drug delivery, resulting in improved efficacy and safety [4]. Other proposed new therapeutic approaches have been the use of probiotics [5] or taking advantage of the anti-inflammatory properties of polyphenols [6]. More recently, oral drugs capable of inhibiting Janus kinases (JAKs)—termed targeted synthetic DMARDs (tsDMARDs) or, more simply, JAK inhibitors (JAKis)—have been introduced for RA therapy [7]. JAKis prevent the action of proinflammatory cytokines, such as tumor necrosis factor alpha (TNFα) and interleukin-6 (IL-6), by blocking the JAK/signal transducer and activator of transcription proteins (STAT) transduction pathway after the cytokines interact with their receptors [8,9]. The first JAKi approved by the Food and Drug Administration (FDA) for the treatment of RA was tofacitinib (TOFA) about a decade ago [10]. Following the FDA approval, TOFA was also approved by the European Medicines Agency [11,12]. TOFA preferentially inhibits JAK1 and JAK3 [13], with consequent inhibition of the activity of several pro-inflammatory cytokines that are produced by CD4+ T cells and synovial fibroblasts, including IL-6, and it has been shown to be effective in several models of experimental arthritis [14,15].

The potential role of TOFAs in the treatment of RA also emerged from the results of several studies that analyzed the best treatment strategy for RA patients who were refractory to initial second-line therapy with anti-tumor necrosis factor-α (TNFα) bDMARDs. These studies concluded in most cases that switching to a second anti-TNFα bDMARD was not the best strategy, but rather it was advisable to switch to a biologic with a different mechanism of action [16]. With the approval of JAKis, inhibition of multiple cytokines, including IL-6, seems to be a very attractive strategy and is indicated for the treatment of moderate-to-severe active RA in adult patients who have responded inadequately or are intolerant to one or more csDMARDs or bDMARDs [17].

One of the biggest advantages of TOFA over bDMARDs is that it is a synthetic small molecule that is, therefore, non-immunogenic. Conversely, the significant immunogenicity of biologics, due to their protein nature, may be responsible for the formation of anti-drug antibodies that may limit their long-term activity [18,19]. Although TOFA, thus, has the potential to be maintained in therapy for a long period of time, to date, definitive data on its retention rate in the real world are still lacking. The main purpose of this retrospective multicenter study was to analyze the rate of TOFA maintenance over a long period of time, up to 48 months, in a large cohort of RA patients referred to tertiary rheumatology centers in Italy. The secondary objectives of the study were to analyze potential risk factors predictive of therapy discontinuation and to identify adverse events that occurred during treatment.

## 2. Materials and Methods

### 2.1. Data Source and Collection

This multicenter retrospective cohort study involved 23 tertiary referral rheumatology centers in Italy. The research is part of the BIRRA (BIologics Retention Rate Assessment) project, aimed at studying the retention of therapy of innovative antirheumatic drugs over time. The study was approved by the ethics committees of all participating centers and was conducted in accordance with the Declaration of Helsinki and good clinical practice guidelines. All participants provided written consent to participate in the research. Patient data were extracted from the clinical databases of individual centers participating in the study. The diagnosis of RA was made according to the 2010 American College of Rheumatology (ACR)/European Alliance of Associations for Rheumatology (EULAR) criteria [20]. All patients were aged ≥18 years and were treated with TOFA as monotherapy or in combination with csDMARD, with or without the addition of steroids. The patient cohort was followed for 48 months, from April 2019 to April 2023. Demographic characteristics; smoking habit; previous and ongoing treatments; comorbidities; and laboratory data, including positivity for rheumatoid factor (RF) and anti-citrullinated protein antibody (ACPA), were recorded. Comorbidities considered were diabetes, dyslipidemia, history of major adverse cardiovascular events (MACEs), cancer, and hypertension. Disease activity was assessed in all patients by calculating the disease activity score 28-ESR (DAS28-ESR). The TOFA retention rate and any reasons for discontinuation of therapy were then analyzed.

### 2.2. Statistical Analysis

The median and interquartile range (IQR) were calculated for variables with nonparametric distributions. Categorical data were expressed as number and percentage. Cox regression analysis was used to identify predictors of TOFA discontinuation. These data were presented as the hazard ratio (HR) and the corresponding 95% confidence interval (CI). The TOFA retention rate curve was constructed using the Kaplan–Meier method. A *p* value ≤ 0.05 was considered statistically significant. All statistical analyses were two-sided and were performed using Jamovi statistical software version 2.3 (http://www.janovi.org, last visit 31 May 2023).

## 3. Results

### 3.1. Demographic and Clinical Characteristics of Patients

A total of 171 of the 213 patients included in the study were female (80%). The median age at the start of follow-up was 60 years (IQR: 51–67). Overall, 44 (20.6%) patients were active smokers, and 41 (19.2%) had a history of previous smoking. The median body mass index (BMI) was 24.8 (IQR: 22–27.6). The median disease duration was 73 months (IQR: 22–146). Positivity for RF was present in 140 (65.7%) patients and for ACPA in 131 (61.5%) patients. One hundred twenty patients were positive for both RF and ACPA (56.3%). Measurement of pain intensity using the visual analog scale (VAS) expressed with a score from 0 to 100 showed a median value of 70 (IQR: 50–80). The median DAS28-ESR at baseline was 5.34 (IQR: 4.64–5.97). TOFA therapy was associated with methotrexate (MTX) in 163 (76.5%), leflunomide in 52 (24.4%), sulfasalazine in 27 (12.7%), and hydroxychloroquine in 80 (37.6%) patients. Previous biologic therapy had been with anti-TNFα in 122 (52.8%), anti-IL6 receptor in 49 (23%), IL-1 receptor antagonist in 3 (1.4%), anti-CD20 in 18 (8.5%), and CD80/CD86 blocker in 38 (17.8%) patients. In total, 151 of the 213 patients (56.8%) included in this study had received at least one bDMARD, and in 55 patients (25.8%), TOFA administration had been preceded by treatment with one or more JAKis. The most common comorbidity observed in our case series was hypertension, present in 61 (28.6%) patients. The other comorbidities observed most often were dyslipidemia (39 patients, 28.6%) and diabetes (12 patients, 5.5%). MACEs were present in 9 (4.2%) patients, and a history of previous malignancy was detectable in 11 (5.2%) patients. The demographic and clinical characteristics of the patients studied are summarized in Table 1.

### 3.2. TOFA Survival in Therapy

The retention rate of TOFA was analyzed in the case series considered here. The Kaplan–Meier curve of the cumulative probability of the retention rate over a 48-month treatment period is shown in Figure 1. At month 12, the treatment retention rate was 86.5% (95% CI: 81.8–91.5%). At months 24, 36, and 48, the probability that patients were still on TOFA treatment was 78.8% (95% CI: 78.8–85.2%), 63.8% (95% CI: 55.1–73.8%), and 59.9% (95% CI: 55.1–73.8%), respectively. These data are shown below the curve graph in tabular form, along with the number of individuals at risk and the number of events observed at the different time points considered.

### 3.3. Predictive Factors of TOFA Discontinuation

Predictive independent variables affecting the TOFA retention rate were analyzed using Cox regression and expressed numerically as the HR and 95% CI. Their value represents the probability of treatment discontinuation in the presence of the predictive factor analyzed. The HR of male sex for TOFA therapy discontinuation was 0.85 (95% CI: 0.23–3.14, *p* = 0.81). Other predictors analyzed were smoking habit (HR 0.28, 95% CI: 0.05–1.46, *p* = 0.31), positivity for RF (HR 1.76, 95% CI: 0.33–9.29, *p* = 0.50) or ACPA (HR 0.26, 95% CI: 0.05–1.38, *p* = 0.11), presence of diabetes (HR 0.32, 95% CI: 0.03–3.47, *p* = 0.34), hypertension (HR 0.36, 95% CI: 0.1–1.37, *p* = 0.13), history of MACE (HR 3.31, 95% CI: 0.57–19.33, *p* = 0.18), previous malignancy (HR 1.44, 95% CI: 0.12–17.94, *p* = 0.77), concomitant csDMARD therapy (HR 1.13, 95% CI: 0.41–3.15, *p* = 0.80), previous therapy with a bDMARD (HR 5.13, 95% CI: 0.84–31.23, *p* = 0.07) or with more than one bDMARD (HR 1.67, 95% CI: 0.32–8.81, *p* = 0.54), age (HR 1.03, 95% CI: 0.97–1.10, *p* = 0.38), disease duration (HR 1.44, 95% CI: 0.12–17.94, *p* = 0.77), concomitant steroid use (HR 1.11, 95% CI: 0.99–1.25, *p* = 0.78), DAS28-ESR (HR 0.71, 95% CI: 0.44–1.16, *p* = 0.17), and BMI (HR 0.96, 95% CI: 0.82–1.13, *p* = 0.61). These data are summarized in Figure 2.

### 3.4. Reasons for Discontinuation of TOFA

TOFA discontinuation was due to lack of clinical response in 37 cases (17.3%). Specifically, primary failure was observed in 21 cases (9.8%) and secondary failure in 16 cases (7.5%). The main reasons for treatment interruption due to adverse events (AEs) were the occurrence of tumors during therapy in two cases (0.9%) and nonspecific gastrointestinal symptoms in two cases (0.9%). Other less frequent cases (one case each, 0.4%) were the occurrence of uveitis, serum levels of creatine phosphokinase or alanine aminotransferase above the normal range, and hypertension. In four cases (1.8%), the cause of discontinuation of TOFA could not be determined. Multiple regression analysis was not possible because the number of cases was too small. These data have been summarized in Table 2.

## 4. Discussion

In this study, we reported the TOFA retention rate in patients with RA through analysis of real-world data obtained from 23 tertiary referral rheumatology centers in Italy up to 48 months of treatment. We found that at 12 months after the start of therapy, the probability of maintaining TOFA was 86.5%. At months 24, 36, and 48, the probability that patients were still on TOFA therapy was 78.8%, 63.8%, and 59.9%, respectively.

Several studies have demonstrated the efficacy of TOFA in the treatment of RA, both in prospective randomized trials and from retrospective analysis of data obtained in clinical practice [21,22,23]. In most studies, the efficacy of TOFA in monotherapy was not significantly different from its use in combination with MTX [24].

The study of the retention rate of drugs used as second-line for the treatment of rheumatoid arthritis is highly variable and often depends on the type of drug considered [25,26]. The main factors influencing the retention rate are efficacy and safety. Given the chronic nature of RA, the availability of a drug that can persist in therapy for prolonged periods of time is essential for good control of the disease, but also to allow a good quality of life for the patient. When a drug fails, changes in therapy are made that progressively reduce its effectiveness [27,28,29].

The results obtained from our study showed a better retention rate than that reported by Iwamoto et al., who observed a TOFA retention rate of 76.4% after only 24 weeks of therapy [30]. Several other studies have reported a lower rate of TOFA therapy retention than our research. Pope et al. reported TOFA retention rates of 62.7% and 49.6% after 12 and 24 months of treatment, respectively [31]. In a very large study describing the prescription and maintenance of treatment with molecularly targeted drugs for RA, TOFA retention rates of 69, 58, and 50% were observed after 1, 2, and 3 years, respectively [32]. It has been reported from a Turkish study, in which the TOFA retention rate was 63.9% at 1 year [33]. In another study, Movahedi et al. found a drug retention rate of 63.3% at a mean follow-up of 23.2 months [34].

The higher retention rate in TOFA therapy reported in our study may have several explanations, including genetic factors, the mode of patient enrollment, and rheumatologist’s propensity to change therapy. We then showed that none of the potential predictors of therapy discontinuation considered, including patient age, previous treatment with biologics, concomitant use of csDMARD, and positivity for RF and/or ACPA, were statistically significant after regression analysis. Similar data were reported by Bilgin et al. [33], Movahedi et al. [34], Bird et al. [35], and Finckh et al. [36], who found no relevant predicting factor for TOFA discontinuation. Our results are also in agreement with a recently published paper analyzing data from a Korean registry, in which no predictive factors for discontinuation of therapy were found, except RF and ACPA positivity, both of which were associated with higher drug retention rates [37]. In contrast, in another study, positivity for ACPA was reported as a risk factor for TOFA discontinuation [38], showing that the role of seropositivity in the retention of TOFA therapy is still far from being fully elucidated.

Other factors predicting the rate of TOFA retention rate have been identified in some studies. In the study from Shouval et al. [39], the TOFA therapy retention rate was correlated with the number of treatment lines, early introduction into therapy being associated with longer drug survival. In the study by Pope et al. [40], an increased risk of discontinuation was associated with age ≥ 56 years and inadequate response to anti-TNFα bDMARDs. In some studies, it has been reported that the TOFA retention rate was influenced by previous therapy with bDMARDs, including the report by Ebina et al., who found a higher retention rate in bDMARD-naïve as compared with bDMARD-switched patients [25]. In the present study, we found an excellent safety profile of TOFA. In fact, the main cause of treatment discontinuation was primary or secondary ineffectiveness.

One of the main problems with rheumatoid arthritis therapies is safety. In fact, therapies aimed at reducing inflammation—and, consequently, the effectiveness of the response against pathogens—may expose patients to potentially increased infectious risk. In addition, it is possible that drugs that act by inhibiting the JAK/STAT pathway, such as drugs of the JAKis class, may result in effects unrelated to anti-inflammatory function due to their pleiotropic action. Regarding the risk of infection, some retrospective studies have shown an equal risk between patients treated with TOFA when compared with either csDMARDs or bDMARDs plus MTX [22]. Further analysis of multiple databases reported TOFA to be related to a higher infectious risk only when compared with etanercept, while it had a comparable risk profile to adalimumab, golimumab, tocilizumab, and abatacept [41]. In a Japanese observational study using a large national database, a herpes zoster incidence rate of 7 was reported in TOFA-treated patients compared with 2.4 in biologic-treated patients [42]. In the US Corrona RA registry, an increased risk of herpes zoster infection in patients treated with TOFA was also observed. However, these infections were non-serious. There were no warning signs for other types of infection [24].

An additional cause for concern is the possible occurrence of neoplasms due to decreased anti-tumor immunosurveillance. In a prospective analysis of the same US Corrona RA registry, the rate of occurrence of neoplasms was comparable to that of bDMARDs [43]. A meta-analysis confirmed this evidence [44]. Finally, an important concern that arose with the use of TOFAs was the possible increased incidence of thromboembolic episodes and cardiovascular disease. In a retrospective study, no increased risk of cardiovascular disease was observed with TOFA at the standard dose of 5 mg twice daily [45]. It is interesting to note that an increased cardiovascular risk is instead associated with corticosteroid use. The use of alternative drugs to steroids, including TOFA, may therefore be indirectly responsible for reducing cardiovascular risk. The risk of venous thromboembolism was also found to be comparable between TOFAs and bDMARDs [46]. However, new JAKIs that show better efficacy and safety need to be found in the near future. In this regard, in silico studies represent an interesting research frontier [47].

In our study, side effects accounted for only 3.75% of TOFA discontinuation cases. This result is in line with data reported by others, including the study by Mori et al., in which AEs were responsible for 4% of TOFA discontinuations [48]. However, other authors have reported a significantly higher incidence of AE as a reason for TOFA discontinuation. For example, Pope et al. reported that TOFA discontinuation due to AEs was 26.9% [31]. Ebina et al. found an all-causality AEs rate of 38.5% [25], Mueller et al. reported an incidence of AEs during therapy of 23.6% that led to TOFA dropout [49], and Bilgin et al. reported that TOFA therapy was discontinued due to AEs in 15% of cases. In this last study, most AEs involved allergic skin reactions [33].

Importantly, cases of serious herpes zoster (HZ), defined as a disseminated disease that caused multiple skin lesions and/or damage to the central nervous system, lungs, liver, or kidneys, were not observed. HZ was also considered serious when it caused complicated eye disease [50]. Mild forms of HZ presenting as a self-limiting vesicular rash were not considered in our study. Since the recombinant vaccine has been available in Italy since 2021, and the 2019 ACR/EULAR recommendations suggest vaccination before the start of therapy [51], none of the patients included in the study were vaccinated.

This result is partially in contrast to what has been previously observed in phase II and III studies [52,53]. However, it has been observed that this complication during TOFA therapy is more frequent in geographic areas where HZ infection has a high prevalence, such as regions in East Asia, or if certain specific genetic variants are present in treated patients [54]. Another important AE observed during several studies is the increased incidence of thromboembolic episodes [55]. No cases of MACE were recorded in our study. Many factors related to both the patient and the treating physician may, therefore, account for the discrepancies in the results reported by the various studies.

Although the safety of TOFA has been the subject of a large surveillance study, its design revealed several methodological limitations [56]. Nevertheless, regulatory agencies have issued the same black boxes restricting the use of TOFA and other JAKis in the presence of selected risk factors, but imitations on the ability to prescribe TOFA have led to intense debate among rheumatologists [57]. It is likely that in the coming time, the safety profile of TOFA and other JAKis approved for the treatment of RA will be definitively clarified.

Regarding the retention rate of TOFA therapy, it should be remembered that in some nations, TOFA is still available in a modified-release, 11 mg, once-daily (QD) formulation, in addition to the 5 mg twice-daily (BID) formulation [58]. In an interesting study, the QD formulation was shown to increase the retention rate of therapy compared with the BID formulation [59].

The limitations of our study are mainly related to its retrospective design, which may have resulted in a selective bias toward patients. In addition, since it was a study involving several Italian rheumatology centers, it may have included patients with different clinical histories or genetic backgrounds. Also, from our data, it was not possible to perform an analysis of the TOFA retention rate by comparing patients in different lines of therapy. Large prospective controlled studies are needed to clarify these issues.

## 5. Conclusions

In conclusion, our study showed that TOFA has a relatively long therapeutic retention rate in patients with RA, evaluated up to 48 months of therapy. This study extends the results obtained in experimental studies to a real-world condition. Further studies are needed to confirm these results and establish the safety of long-term treatment, as well as to compare the retention rate of TOFA with that of bDMARDs or other JAKis.

## Figures and Tables

**Figure 1 medicina-59-01480-f001:**
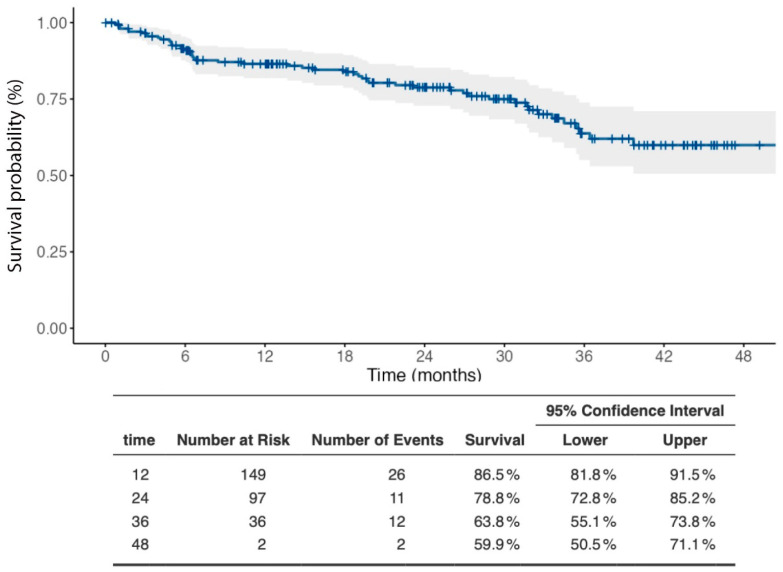
Kaplan–Meier curve showing retention rate of TOFA in RA patients over 48 months of follow-up. Below the curve, the number of patients still on treatment (number at risk) in the different observation periods, the number of treatments discontinued (number of events), and the probability of remaining on treatment (survival) in the various time intervals considered, with their 95% confidence intervals.

**Figure 2 medicina-59-01480-f002:**
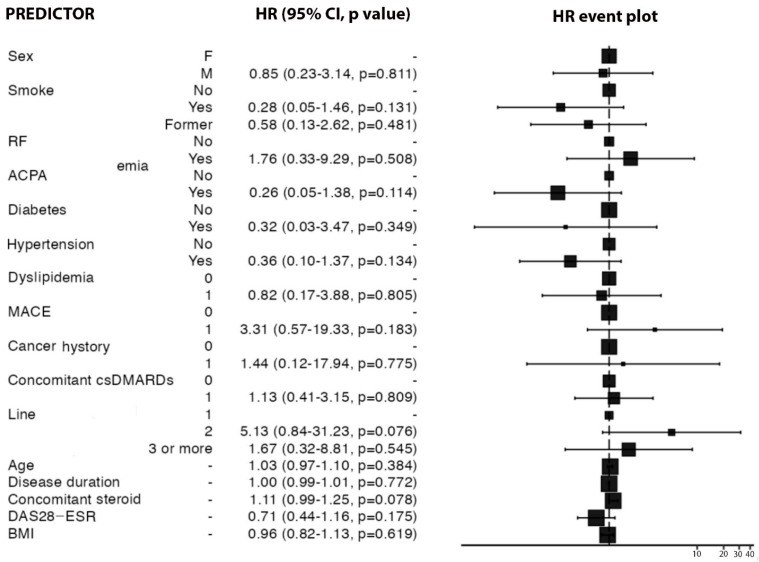
Analysis of predictive factors of TOFA retention rate. On the left side of the figure are listed the independent factors potentially predictive of TOFA therapy retention rate, with the respective hazard ratios (HRs), 95% confidence intervals, and statistical significance expressed as *p* value. On the right, the results are represented graphically by HR event plot.

**Table 1 medicina-59-01480-t001:** Demographic and clinical characteristics of the patients studied.

Characteristic	
Age, year, median (IQR)	60 (51–67)
Sex, n. (%)	
Female	171 (80.2%)
Male	42 (17.8%)
BMI, Kg/m^2^, median (IQR)	24.8 (22–27.6)
Smoker, n (%)	
Yes	44 (20.7)
Former	41 (19.2)
No	109 (51.2)
Disease duration, months, median (IQR)	73 (22–146)
RF positive, n (%)	140 (65.7)
ACPA positive, n (%)	131 (61.5)
ESR, mm/h, median (IQR)	32 (20–50)
CRP, mg/dL, median (IQR)	1.4 (0.5–3.3)
VAS, 0–100, median (IQR)	70 (50–80)
DAS28-ESR, median (IQR)	5.34 (4.64–5.97)
Line of treatment, n, median (IQR)	2 (1–3)
Concomitant csDMARDs, n (%)	
MTX	163 (76.5)
LFN	52 (24.4)
SSZ	27 (12.7)
HCQ	80 (37.6)
Steroid (PDN-Eq) dose, mg/die, median (IQR)	50 (5–5)
Previous usage of bDMARDs, n (%)	
Anti-TNFα	112 (52.6)
Anti-IL6R	49 (23)
IL1ra	3 (1.4)
Anti-CD20	18 (8.5)
CD80/CD86 inhibitor	38 (17.8)
Previous usage of tsDMARDs, n (%)	
One	52 (24.4)
Two or more	3 (1.4)
Comorbidities, n (%)	
Diabetes	12 (12.5)
Dyslipidemia	39 (18.3)
Previous MACE	9 (4.2)
Hypertension	61 (28.6)
History of cancer	11 (5.2)

IQR = interquartile range; BMI = body mass index; RF = rheumatoid factor; ACPA = anti-citrullinated protein antibody; ESR = erythrocyte sedimentation rate; CRP = C reactive protein; VAS = visual analogue scale; DAS28-ESR = disease activity score 28-ESR; csDMARDs = conventional synthetic disease-modifying anti-rheumatic drugs; PDN-Eq = prednisone equivalent; bDMARDs = biological DMARDs; tsDMARDs = targeted synthetic DMARDs; MTX = methotrexate; LFN = leflunomide; SSZ = sulfasalazine; HCQ = hydroxychloroquine; TNF = tumor necrosis factor; IL6R = interleukin-6 receptor; IL1ra = interleukin-1 receptor antagonist; MACE = major adverse cardiac event.

**Table 2 medicina-59-01480-t002:** Causes of discontinuation of TOFA therapy.

No. of Cases (%)	
Primary failure	21 (9.8%)
Secondary failure	16 (7.5%)
Cancer	2 (0.9%)
Non-specific gastrointestinal disorders	2 (0.9%)
Hypertension	1 (0.4%)
Uveitis	1 (0.4%)
HyperCKemia	1 (0.4%)
Elevated serum ALT levels	1 (0.4%)
Unknown	4 (1.8)

HyperCKemia = elevated serum creatine kinase levels; ALT = alanine aminotransferase.

## Data Availability

All data generated or analyzed during this study are included in this published article.

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
