# Peer review of "Long-Term Retention Rate of Tofacitinib in Rheumatoid Arthritis: An Italian Multicenter Retrospective Cohort Study"

_medicina, 2023, doi:10.3390/medicina59081480_

Round 1
Reviewer 1 Report
As ref 28, Mori mentioned the improvement of RA was observed 68 cases, and not the retention rate and the number of the patiets is only 100. As for Japanese data for long TOFA's retention rate, I recommend you to quote big data including more than 1000 cases (Trend in prescription and treatment retention of molecular-targeted drugs in 121,131 Japanese patients with rheumatoid arthritis: A population-based real-world study. Rheumatol. 2022 Aug 20;32(5):857-865) They showed three years survival (retention rate) of TOFA in all Japanese cases, that is 69,58,50% respectively in table 1).
As for Herpes Zoster, no case was reported in your survey, which is actually incredible. You should mention about vaccination in your cases. In Japan IR is 7.0/100person years in the patients treated with TOFA in whole country and median event occrred thirteen months (Mod Rheumatol. 2022 Oct 29:roac133. doi: 10.1093/mr/roac133), Thus it seems weired that no case of Herpes zoster was reported during your long survey more than 4 years even if taking the difference of races.
at the top of page 7, same sentece was repeated.
At month 12, the treatment retention rate was 86.5% (95% CI: 81.8- 91.5%).
Author Response
Responses to Reviewer 1
(Changes to the text have been highlighted in green)
- As ref 28, Mori mentioned the improvement of RA was observed 68 cases, and not the retention rate and the number of the patients is only 100. As for Japanese data for long TOFA's retention rate, I recommend you to quote big data including more than 1000 cases (Trend in prescription and treatment retention of molecular-targeted drugs in 121,131 Japanese patients with rheumatoid arthritis: A population-based real-world study. Mod Rheumatol. 2022 Aug 20;32(5):857-865) They showed three years survival (retention rate) of TOFA in all Japanese cases, that is 69,58,50% respectively in table 1). Answer: The reference to the study by Mori et al. has been removed from the text. In its place, the study by Takabayashi et al. has been included as suggested.
- As for Herpes Zoster, no case was reported in your survey, which is actually incredible. You should mention about vaccination in your cases. In Japan IR is 7.0/100person years in the patients treated with TOFA in whole country and median event occurred thirteen months (Mod Rheumatol. 2022 Oct 29:roac133. doi: 10.1093/mr/roac133), Thus it seems weird that no case of Herpes zoster was reported during your long survey more than 4 years even if taking the difference of races. Answer: We thank the reviewer for this observation and for the citation that was included in the text. We considered only serious cases of HZ. Mild forms were not considered in our study. This has now been specified in the manuscript. Because the recombinant vaccine has been available in Italy since 2021 and the 2019 ACR/EULAR recommendations suggest vaccination before starting therapy, none of the patients included in the study were vaccinated. This point was also added to the text.
Reviewer 2 Report
The present article evaluates retrospectively, but multicentric the long-term retention rate of tofacitinib in rheumatoid arthritis. The topic is relevant, but certain deficiencies identified in both content and form need to be addressed based on the specific recommendations below:
1. Abbreviations should be explained in the first sentence, then only used in abbreviated form (e.g. RA etc.), but if they are mentioned only once, the unabbreviated form should be used. Please revise the basis on these principles throughout the manuscript since the abstract is treated differently from the main text in this respect.
2.RA not AR therapy in the introduction section near [4].
3. In no case should bibliographic clues [x] be inserted after the point. Please review and see [15,16].
4. It is important to discuss and unmet needs of current JAKis, hence further studies to improve efficacy and safety profiles (especially thromboembolic risk), computational studies are essential. I suggest you check and consult: PMID: 37375255.
5. The introduction section should be more detailed, as it is slightly lacking in contrast to the complexity of the topic evaluated, so current and relevant information on the need to study nanoapplications to address unmet needs in all DMARDs (cs, b and tsDMARDs) should be included. It is important to understand why nanoapplications are needed and what problems they could solve, as they also have drawbacks. I suggest you check and refer to: PMID: 37031724.
6.The scope of the paper should be improved in terms of describing the contribution to the field under review and the elements of scientific novelty presented, especially as this is not the first manuscript to evaluate these aspects.
7. Besides JAKis, adjuvant therapies such as polyphenols and or microbiota regulators are very important to mention as they have been shown to improve the management of RA patients. I suggest you check and consult: PMID: 34684377 and PMID: 33998910.
8.The concluding part of the abstract should be improved in terms of the results and what future research directions this research can refer to.
Author Response
Responses to Reviewer 2
(Changes to the text have been highlighted in yellow)
- Abbreviations should be explained in the first sentence, then only used in abbreviated form (e.g., RA etc.), but if they are mentioned only once, the unabbreviated form should be used. Please revise the basis on these principles throughout the manuscript since the abstract is treated differently from the main text in this respect. Answer: Abbreviations were carefully checked both in the abstract and in the main text. They were removed if the abbreviated word was not repeated in the text. Moreover, some unexplained abbreviations have been explained in full.
2.RA not AR therapy in the introduction section near [4]. Answer: AR was corrected in RA
- In no case should bibliographic clues [x] be inserted after the point. Please review and see [15,16]. Answer: All points before bibliographic clues have been removed from the text.
- It is important to discuss and unmet needs of current JAKis, hence further studies to improve efficacy and safety profiles (especially thromboembolic risk), computational studies are essential. I suggest you check and consult: PMID: 37375255. Answer: A comment was added in the discussion section on the usefulness of in-silico studies to find new, more effective and safer JAKis. The suggested literature reference has been added.
- The introduction section should be more detailed, as it is slightly lacking in contrast to the complexity of the topic evaluated, so current and relevant information on the need to study nanoapplications to address unmet needs in all DMARDs (cs, b and tsDMARDs) should be included. It is important to understand why nanoapplications are needed and what problems they could solve, as they also have drawbacks. I suggest you check and refer to: PMID: 37031724. Answer: A sentence was added in the introduction section on the usefulness of the nanomedical approach to improve drug delivery and consequently efficacy and safety in RA. The suggested citation has been added.
6.The scope of the paper should be improved in terms of describing the contribution to the field under review and the elements of scientific novelty presented, especially as this is not the first manuscript to evaluate these aspects. Answer: It was better emphasized in both the abstract and the introduction that the peculiarity of the work is in the length of follow-up, which is up to 48 months.
- Besides JAKis, adjuvant therapies such as polyphenols and or microbiota regulators are very important to mention as they have been shown to improve the management of RA patients. I suggest you check and consult: PMID: 34684377 and PMID: 33998910. Answer: The potential use of new therapeutic tools, such as the use of probiotics or polyphenols, has been added in the introduction, and suggested citations have been added to the text.
8.The concluding part of the abstract should be improved in terms of the results and what future research directions this research can refer to. Answer: In the conclusions of the abstract, we added that more large, long-term observational studies are urgently needed to confirm these findings.
Round 2
Reviewer 2 Report
The authors have significantly improved the manuscript based on the suggestions received.